# Associated factors of pelvic organ prolapse among patients at Public Hospitals of Southern Ethiopia: A case-control study design

Asfaw Borsamo[1]*, Mohammed Oumer[2], Ayanaw Worku[2], Yared Asmare[2]

1 Department of Human Anatomy, College of Medicine and Health Sciences, Hawassa University, Hawassa, Ethiopia, 2 Department of Human Anatomy, College of Medicine and Health Sciences, University of Gondar, Gondar, Ethiopia

* aseborsa@gmail.com

**Data Availability Statement:** All relevant data are within the manuscript and its Supporting Information files.

## Abstract

### Background

Pelvic organ prolapse (POP) is the descent of the vaginal wall, cervix, uterus, bladder, and rectum downward into the vaginal canal. It occurs owing to the weakness of the structures supporting and keeping pelvic organs in anatomic position. Prolapse occurs due to exposure to risk factors; women in developing countries are highly predisposed to the risk factors of the prolapse. No study assesses POP in Southern Ethiopia.

### Methods

A case-control study design was employed in 369 participants (123 cases and 246 controls) of seven randomly selected Public Hospitals of Southern Ethiopia from February-June, 2020, using a structured questionnaire. All patients diagnosed with prolapse (stage I- IV) were included as cases; patients free of prolapse (stage 0) were taken as controls after physicians had performed a diagnosis and vaginal examination. Bivariate and multivariable logistic regression analyses were performed using SPSS.

### Results

In this study, after adjusting for covariates, age of the women $\geq$ 45 years (AOR = 5.33, 95% CI: 1.47, 9.05), underweight (AOR= 4.54, 95% CI: 1.4, 15.76), low income (AOR = 2.5, 95% CI:1.14, 5.59), parity $\geq$5 (AOR = 5.2, 95% CI: 2.2, 12.55), assisted vaginal delivery (AOR= 4, 95% CI: 1.55, 11.63), instrumental delivery (AOR= 3.5, 95% CI:1.45, 84), sphincter damage and vaginal tear (AOR = 3.2, 95% CI:1.44,7.1), carrying heavy loads (AOR= 2.5, 95% CI:1.2, 5.35), and prolonged labor $\geq$24 hours (AOR = 3.3, 95% CI:1.12, 97) were significant associated factors of prolapse. The odds of developing prolapse is lower among women attended school. Most(84.55%) of the women with prolapse were delayed for the treatments and only surgical interventions were done as treatments. Most of them claimed lack of social

**Funding:** Mr. Asfaw Borsamo took award From University of Gondar with grant number 045028/2020. The website of University of Gondar is http://www.uog.edu.et/en/. The University of Gondar had no role in study design, data collection and analysis, decision to publish, or preparation of the manuscript.

**Competing interests:** The authors have declared that no competing interests exist.

**Abbreviations:** AOR, Adjusted Odds Ratios; BMI, Body Mass Index; CI, Confidence Intervals; COR, Crude Odds Ratios; CS, Cesarean Section; HMIS, Health Management Information System; Kg, Kilogram; IAP, Intra-abdominal Pressure; NGOs, Non-Governmental Organization; OPD, Outpatient Department; POP, Pelvic Organ Prolapse; SNNPR, South Nations Nationalities, Peoples' Region; UAE, United Arab Emirates; USA, United State of America; UVP, Uterovaginal Prolapse.

support, lack of money, and social stigma as the main reasons for the delay in seeking treatments.

## Conclusions

Older age, low educational status, underweight, low income, higher parity, assisted vaginal delivery, prolonged labor, sphincter damage, and carrying heavy loads were significant associated factors of POP. It is better to screen older age women by doing campaigns against the prolapse. Also, responsible bodies should work on raising awareness of women as well as awareness of the community about the prolapse through expanding health education. Moreover, informing women to practice pelvic muscle training daily, raising women's income to empower them, and help of family members to reduce carrying an overload of mothers are recommended.

## Introduction

Pelvic organ prolapse (POP) is defined as the herniation or descent of the vaginal wall, cervix, uterus, bladder, and rectum downward along the vaginal lumen [1]. Universally, thirty percent of the women who have delivered a child are affected by POP resulting in debilitating morbidity [2] The magnitude of POP is very variable depending on the level of the development of the nations and the level of the exposure of the women to the risk factors [3]. Globally, the magnitude of POP is highly variable ranging between 3% and 64.6% [4, 5]. The prevalence of POP was 15.6% in Bangladesh [6], 9.1% in China [7], 48% in Australia [8], 29.6% in the United Arab Emirates (UAE) [9], 52% in Brazil [10], and 15% in Nepal [2]. In Africa, the prevalence of POP in Tanzania was 64.6% [5], in Egypt was 19.4% [11], in Nigeria was 6.5% [12], and in the Gambia was 46% [13]. Similarly, the prevalence of the POP in Ethiopia is highly variable. The prevalence rate in Jimma hospital, Saint Paulos Hospital, in Dabat community, in Kersa community, and Benchi Maji Zone was 40.7% [14], 15% [15], 56.4% [16], 20.9% [17] and 13.3% [18], respectively.

The risks of POP are categorized into three. The leading risk factors for POP are obstetric factors [16, 19]. It includes a higher number of children (vaginal delivery), age of the pregnancy, prolonged labor, the assistance of non-professional personnel, instrumental delivery, home delivery, age of the first pregnancy, and immediate return to work following delivery [19, 20]. The second risk factor category for POP are demographic and socio-economic factors including advanced age (the most important), body mass index (BMI), ethnicity, the family history of POP, low income, and low literacy level [20]. The third categories of risk factors for POP are factors that increase intra-abdominal pressure (IAP). These include chronic cough and constipation, carrying and uplifting heavy objects, and engagement in physical labor [7, 8, 16, 17, 19].

In Ethiopia, since it is a very low-income country, the women are highly exposed to the above-mentioned risk factors [16, 17, 21] and these will increase the burden. Literature suggests that in Ethiopia if there is better access to healthcare during pregnancy and delivery, we can reduce morbidity later in life. Moreover, the prolapse can be reduced if there is better access to drinking water, transportation of agricultural products to and from the markets, electricity to avoid carrying heavy loads, and a limited number of children [16, 19, 22]. There is a scarcity of the study assessing associated factors of pelvic organ prolapse in Southern Ethiopia. Exposing the risk factors of POP will initiate the responsible bodies to play a role in the

modification of the factors. Therefore, it was found to be very important to conduct research that assesses the associated factors of POP among patients at the Public Hospitals of South Nations Nationalities and Peoples' Region (SNNPR) of Ethiopia.

## Methods and materials

### Study design, sampling, and study population

A multi-center unmatched case-control study design was conducted from February 28 to June 05, 2020, at Public Health Hospitals of Southern Ethiopia. Out of the total fourteen general and referral public health hospitals in southern Ethiopia, seven hospitals (by taking 50% of the total hospitals) were selected randomly by the simple random sampling method. All available gynecologic patients diagnosed with POP (stage I-IV) at each hospital were taken as cases. To select controls, we used a systematic random sampling method, depending on the gynecologic patient flow of each hospital. Hence, the K value for each hospital varies (Fig 1).

The sample size was calculated by using Epi info version 7 software. For unmatched case-control study design, this software calculates by using double proportion formula [$n = (p_1q_1 + p_2q_2)$

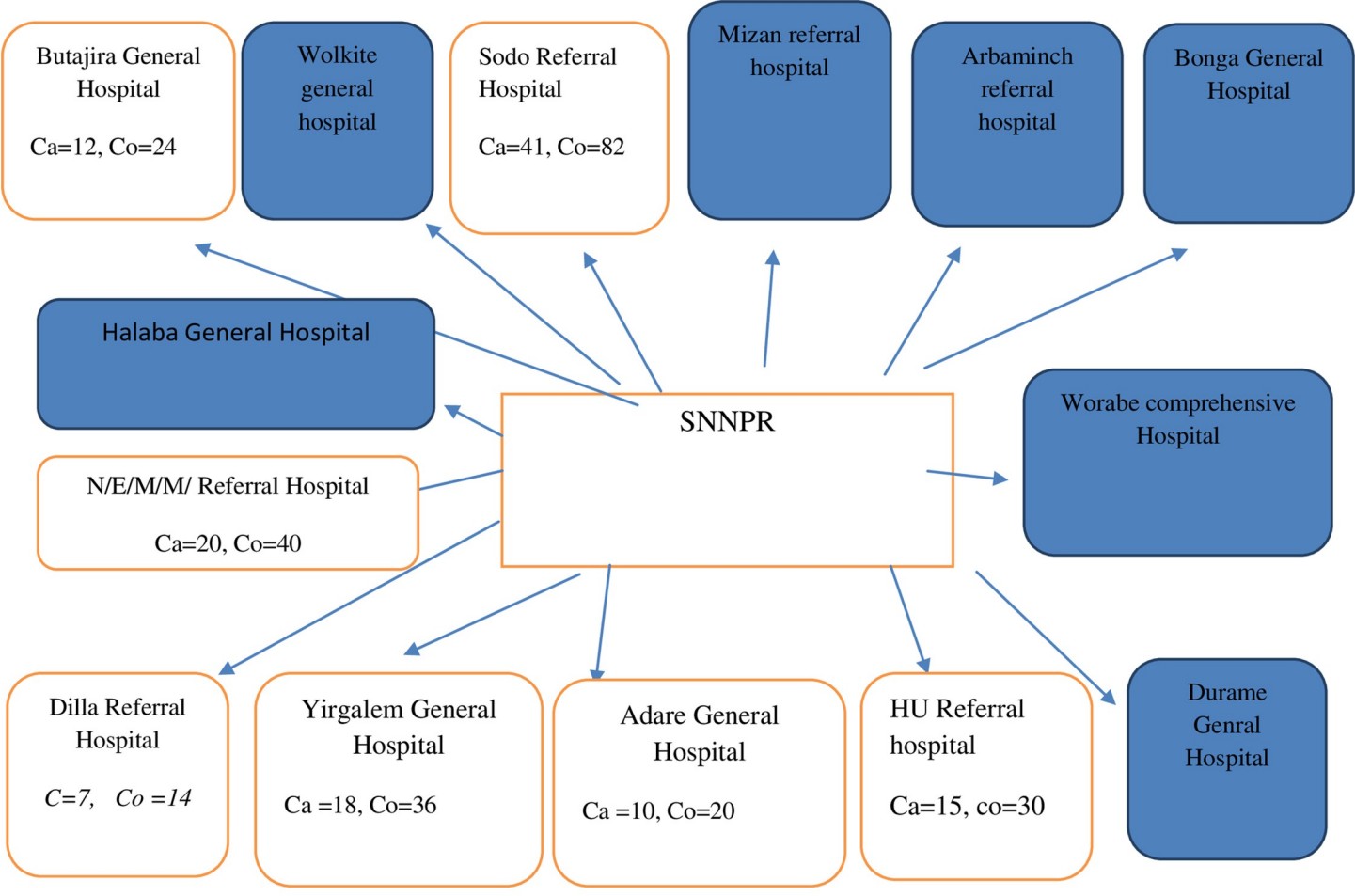

**N/E/M/M/= Negest Elleni Mohammed Memoral, HU = Hawassa University, Ca=cases, Co=controls**

**Fig 1. Sampling procedures.** N/E/M/M/ = Negest Elleni Mohammed Memoral, HU = Hawassa University, Ca = cases, Co = controls.

**Table 1. Sample size calculation.**

| Risk actors | % of case with exposure ($p_2$) | % of control with exposure ($p_1$) | Power | CI (%) | AOR | Sample size |
|---|---|---|---|---|---|---|
| Age >44 | 71.8 | 27.1 | 80 | 95 | 6.8 | 55 |
| Formal education | 85 | 58 | 80 | 95 | 4.3 | 111 |
| Parity | 64.5 | 26.67 | 80 | 95 | 4.5 | 69 |
| Sphincter damage | 15.2 | 2 | 80 | 95 | 8 | 173 |
| Carrying heavy object | 67.4 | 40 | 80 | 95 | 3.1 | 131 |
| Family history of POP | **22.6** | **6.2** | **80** | **95** | **4.9** | **176** |
| Underweight | 34.5 | 14 | 80 | 95 | 3.2 | 164 |
| Delivery assisted by non-health personnel | 63.4 | 39.1 | 80 | 95 | 2.6 | 167 |

$(f(\alpha, \beta)) / ((p_1 - p_2)^2$, n= number of sample size in each group, $\alpha$ = type I error, $\beta$ = type II error]. To calculate the sample size, we used proportions from a similar study conducted in Bahr Dar City, Ethiopia [19]. By using variables of that study, we got a maximum sample size with a variable of 'family history of POP'. The proportion of cases exposed to 'family history of POP' ($p_2$) was 22.6%, and the proportion of controls exposed to 'family history of POP' ($p_2$) was 6.2%. The sample size obtained was 173, after considering the design effect of two, and a 5% non-response rate, the total sample size became 369 patients (123 cases and 246 controls: in 1 to 2 cases to controls ratio). We considered the 95% confidence interval and 80% power (Table 1).

## Measurement and definitions

POP was evaluated and described using a standardized Pelvic Organ Prolapse Quantitative Examination tool. In this technique, the hymen ring (remnant) is considered a reference point [23]. Accordingly, stage 0 refers to no prolapse; stage I: the leading edge prolapse is 1 cm above the level of the hymen (> -1cm); stage II: the leading edge of the prolapse lies through the plane of hymen between -1cm to +1cm; stage III: the most distal portion of the prolapse is between +1cm and +2cm below hymen and stage IV: the complete eversion of the total length of the pelvic organ.

**Cases:-**Those participants who were diagnosed with POP from stage I up to stage IV.
**Controls:-**Those participants who were objectively declared free of POP (stage 0).
**Chronic cough:** Cough that lasts for ≥ 3 weeks since its onset.
**Constipation:** Unsatisfactory defecation characterized by infrequent stool, difficult stool passage, or both at least for the previous 3 months [24].
**Income:** Those who earn less than $1.25/day were considered below the poverty line [24].

## Data collection tools, techniques, and procedures

Data were collected by using the interviewer-guided structured questionnaire. The questionnaire was first prepared in English, and then appropriately translated into the Amharic language. Later, translated back to English to ensure the accuracy of the meaning. During the translation, language experts participated. The questionnaire was developed by the investigators after reviewing different related literature [14, 16, 18, 19]. Then, the questionnaire was evaluated by experts in different professions to ensure validity. A pretest was carried out on 10% of the sample size mainly to examine the validity, approachability, and consistency of the questionnaire. Accordingly, some corrections and modifications were made to the questionnaire. Two midwives collected the data under close supervision by one physician in each hospital. Based on the standardized Pelvic Organ Prolapse Quantitative Examination tool, the stages of POP were determined.

## Data quality control and management

Data quality was controlled through the provision of two-day training to the data collectors and supervisors about the overall objective of the study terms and concepts, the approach of respondents, data collection tools, and techniques of interviewing. The collected data were also cross-checked for its completeness, consistency, accuracy, and clarity daily. The investigators also carried out close supervision during the data collection period to monitor overall data collecting quality.

## Data processing and analysis

After data collection, each questionnaire was again visually checked for completeness, clarity, and accuracy. The data were coded and entered into EPI INFO version 7, and exported into the SPSS version 25 Software. Descriptive, bivariate, and multivariable logistic regression analyses were performed. Selected variables that have a P-value of ≤0.2 at the bivariate analysis were included in the multivariable logistic regression to control all possible confounding factors simultaneously. The goodness of fit was assessed by using the Hosmer and Lemeshow goodness test. Generally, a P-value ≤ 0.05 was considered statistically significant.

## Ethics approval and consent to participate

Ethical clearance was obtained from the Ethical Review Committee of the College of Medicine and Health Sciences, University of Gondar. Permission was obtained from the SNNPR Health Bureau and the administrations of each hospital. A brief explanation of the objectives of the study was given to the participants. Written informed consent was obtained from study participants. Confidentiality was kept carefully by using codes instead of any personal identifiers, and privacy was maintained during the medical examination of each patient.

# Results

## Descriptive data

Total study participants were 369 with a response rate of 100%. Of these, 123 (cases) were clinically diagnosed with POP and 246 were controls (free of POP). Out of 123 POP cases, nearly half (60) of them were stage III, and over one-third (44) were stage IV.

## Socio-demographic characteristics of the participants

Out of 369 participants, one-third (33.3%) were aged ≥45 years. About one-fourth (25.75%) of the participants were from Wolaita and one-fifth from Sidama (20.33%). Over half (58%) of the participants were Protestants. About 44.4% of participants were rural residents and 38.21% were housewives. The majority (65.31%) earn greater than 1,200 birrs monthly. Most (86.4%) of the participants were married. The majority (85.4%) of the participants' BMI value was between 18.5 and 25 kg/m$^2$ (Table 2).

## Obstetric related characteristics of the participants

Of 369 participants, 134(36.3%) gave birth more than four times. One-third (32.8%) of them gave their last birth at home and 44.72% had one or more history of home delivery. About 13.8% of the participants had labor duration extended over ≥ 24 hours at their last delivery. Nearly two-thirds (64.2%) of the participants returned to work resting more than 42 days after the delivery. Most (84.28%, 81.6%, and 85.9%) of the participants' age at marriage, at first delivery, and last delivery was ≥18, ≥20, and ≤40 years, respectively. Around 41.7% of the

**Table 2. Socio-demographic characteristics of the participants at Public Hospitals of Southern Ethiopia, 2020 (N = 369).**

| Variables | | Case (123) | Controls (246) | Total (369) | Percentages % |
|---|---|---|---|---|---|
| **Age** | ≤35 | 13(12.1%) | 94(87.9%) | 107 | 29.00 |
| | 34-44 | 35(25.2%) | 104(74.8%) | 139 | 37.70 |
| | ≥45 | 75(61%) | 48(39%) | 123 | 33.30 |
| **Ethnicity** | Gurage | 16(36.4%) | 28(63.6%) | 44 | 11.90 |
| | Amhara | 8(25.8%) | 23(74.2%) | 31 | 8.40 |
| | Kambata | 7(25.9%) | 20(74.1%) | 27 | 7.30 |
| | Wolaita | 33(34.7%) | 62(65.3%) | 95 | 25.70 |
| | Hadiya | 14(34.1%) | 27(65.9%) | 41 | 11.10 |
| | Sidama | 27(36%) | 48(64%) | 75 | 20.30 |
| | Oromo | 8(346.4%) | 14(63.6%) | 22 | 6.00 |
| | Gamo | 6(25%) | 18(75%) | 24 | 6.50 |
| | Others | 4(40%) | 6(60%) | 10 | 2.70 |
| **Religion** | Orthodox | 43(38.4%) | 69(61.6%) | 112 | 30.40 |
| | Protestants | 68(31.8%) | 146(68.2%) | 214 | 58.00 |
| | Muslim | 9(27.3%) | 24(72.7%) | 33 | 8.90 |
| | Others | 7(70%) | 3(30%) | 10 | 2.70 |
| **Residency** | Rural | 59(36%) | 105(64%) | 164 | 44.40 |
| | Semi-rural | 23(29.9%) | 54(70.1%) | 77 | 20.90 |
| | Urban | 41(32%) | 87(68%) | 128 | 34.70 |
| **Occupational status** | House wife | 88(61.5%) | 53(38.5%) | 141 | 38.21 |
| | Farmer | 23(37.6%) | 39(62.9%) | 62 | 16.81 |
| | Merchant | 23(37.1) | 51(68.9%) | 74 | 20.05 |
| | Employed | 24(26.1%) | 68(73.9%) | 92 | 24.93 |
| **Educational status** | No schooling | 83(63.8%) | 47(36.2%) | 130 | 35.23 |
| | Primary | 18(20.5%) | 70(79.5%) | 88 | 23.85 |
| | Secondary | 15(18.3%) | 67(81.7%) | 82 | 22.22 |
| | Diploma[+] | 7(10.1%) | 62(89.9%) | 69 | 18.70 |
| **Marital status** | Married | 104(32.6%) | 215(67.4%) | 319 | 86.40 |
| | Widowed | 10(32.3%) | 21(67.7%) | 31 | 8.40 |
| | Divorced | 9(47.4%) | 10(52.6%) | 19 | 5.10 |
| **Income/day** | >$1.25 | 38(15.8%) | 203(84.2%) | 241 | 65.31 |
| | ≤$1.25 | 85(66.4%) | 43(33.6%) | 128 | 34.69 |
| BMI (kg/m$^2$) | 18.5-25 | 96(30.5%) | 219(69.5%) | 315 | 85.40 |
| | <18.5 | 21(65.6%) | 11(34.4%) | 32 | 8.67 |
| | >25 | 6(27.3%) | 16(72.7%) | 22 | 5.93 |
| **Family history of POP** | No | 105(32%) | 223(68%) | 328 | 88.89 |
| | Yes | 18(43.9%) | 23(56.1%) | 41 | 11.11 |

participants had one or more history of sphincter damage/vaginal tear and 26.8% of participants had one or more histories of instrumental delivery (Table 3).

## Factors increase intra-abdominal pressures

Out of 369 participants, 162(43.9%) had a history of chronic cough extended for more than three weeks and 68 (18.4%) had a history of chronic constipation extended more than three months. About 41.5% of participants had a history of carrying/uplifting heavy loads (Table 4).

**Table 3. Obstetric related characteristics of the participants at Public Hospitals of Southern Ethiopia, 2020 (N = 369).**

| Variables | | Cases (123) | Controls (246) | Total (369) | Percentage% |
|---|---|---|---|---|---|
| **Parity** | ≤4 | 41(17.4%) | 194(82.6%) | 235 | 63.70 |
| | >4 | 82(61.2%) | 52(38.8%) | 134 | 36.30 |
| Place of the delivery at last delivery | Institution | 78(31.5%) | 170(68.5%) | 248 | 67.20 |
| | Home | 45(37.2%) | 76(62.8%) | 121 | 32.80 |
| **Mode of delivery at last delivery** | SVD | 85(33.6%) | 168(66.4%) | 253 | 68.56 |
| | AVD | 34(49.3%) | 33(50.7%) | 67 | 18.16 |
| | C/SD | 4(91.8%) | 45(8.2%) | 49 | 13.28 |
| **Duration of labor during last delivery (in hours)** | ≤12 | 38(24.8%) | 115(75.2%) | 153 | 41.46 |
| | 13-23 | 56(33.9%) | 109(66.1%) | 165 | 44.72 |
| | ≥24 | 29(56.9%) | 22(43.1%) | 51 | 13.82 |
| **Duration of time to return to work after delivery (in days)** | >42 | 74(31.2%) | 163(68.8%) | 237 | 64.20 |
| | ≤42 | 49(37.1%) | 83(62.9%) | 132 | 35.80 |
| **Delivery interval (in years)** | ≤2 | 77(31.4%) | 168(68.6%) | 245 | 66.40 |
| | >2 | 46(37.1%) | 78(62.9%) | 124 | 33.60 |
| **Age at marriage** | ≥18 | 100(32.2%) | 211(67.8%) | 311 | 84.28 |
| | <18 | 23(39.7%) | 35(60.3%) | 58 | 15.72 |
| **Age at first delivery** | ≥20 | 97(32.2%) | 204(67.8%) | 301 | 81.60 |
| | <20 | 26(38.2%) | 42(61.8%) | 68 | 18.40 |
| **Age at last delivery** | ≤40 | 83(26.2%) | 234(73.8%) | 317 | 85.90 |
| | >40 | 40(76.9% | 12(23.1%) | 52 | 14.10 |
| **History of sphincter damage/vaginal tear** | No | 49(22.8%) | 166(77.2%) | 215 | 58.30 |
| | Yes | 74(48.1%) | 80(51.9%) | 154 | 41.70 |
| **History of instrumental delivery** | No | 72(26.7%) | 198(73.3%) | 270 | 73.20 |
| | Yes | 51(51.5%) | 48(48.5%) | 99 | 26.80 |

## Delay for treatment-seeking

The average delay of the women for the treatments since the onset of the symptoms was 36.41 ± 3.94 months. Out of 123 women with the prolapse, 84.55% delayed the treatments and only surgical interventions were done as treatments. Among 104 women delayed in the treatment of the prolapse, 76%, and 43.9% of them claimed lack of money, and social stigma as the main reasons for the delay, respectively. Out of 104 women delayed for the treatment, 67.48% (83) did not attend formal school and 69.1%(85) of them earn less than $1.25/ day.

## Associated factors of pelvic organ prolapse

On bivariate analysis; age, educational status, BMI, monthly income, family history of POP, parity, mode of delivery, monthly income, age at last delivery, duration of labor, history of

**Table 4. Factors that increase IAP of the participants at Public Hospitals of Southern Ethiopia, 2020 (N = 369).**

| Variables | | Cases (123) | Controls (246) | Total (369) | Percentages % |
|---|---|---|---|---|---|
| **Chronic cough (>3 weeks)** | No | 64(30.9%) | 143(69.1%) | 207 | 56.10 |
| | Yes | 59(34.6%) | 103(63.6%) | 162 | 43.90 |
| **History of chronic constipation (>3 months)** | No | 100(33.2%) | 201(66.8%) | 301 | 81.60 |
| | Yes | 23(33.8%) | 45(66.2%) | 68 | 18.40 |
| **Carrying/uplifting heavy objects** | No | 40(18.5%) | 176(81.5%) | 216 | 58.50 |
| | Yes | 83(54.2%) | 70(45.8%) | 153 | 41.50 |

home delivery, history of sphincter damage/vaginal tear, and carrying/uplifting heavy loads were significant predictors of POP.

Thus, these all variables were put into multivariable logistic regression analysis and age, educational status, BMI, monthly income, parity, mode of delivery, duration of labor, history of sphincter damage/vaginal tear, and carrying/uplifting heavy loads were significant predictors of POP. Accordingly, the women who were aged ≥45 years were about five times [AOR = 5.33(95% CI: 1.47, 9.05)] more likely to develop POP as compared to women who were aged <35 years. Women who attended secondary school 68% [AOR = 0.32(95% CI: 0.11, 0.92)], and diploma (and above) 83% [AOR = 0.17(95% CI: 0.05, 0.57)] were less likely to develop POP as compared with those women who had no schooling at all. The women who had a BMI value ≤ 18.5kg/m$^2$ were 4.5 times [AOR = 4.54 (95% CI: 1.4, 15.76)] more likely to develop POP than women with normal BMI values. Women who earn daily ≤ \$1.25 were 2.5 times [AOR = 2.5(95% CI: 1.137, 5.59)] more likely to acquire POP than their counterparts. The women who had parity ≥5 were five times [AOR = 5.2 (95% CI: 2.2, 12.55)] more likely to develop POP than their counterparts. Women who delivered through assisted vaginal delivery at their last delivery were four times [AOR = 4 (95% CI: 1.55, 11.63)] more likely to develop POP as compared to women delivered through spontaneous vaginal delivery. The women who had prolonged labor (≥24 hours) at the last delivery were three times [AOR = 3.3 (95% CI: 1.12, 9.7)] more likely to develop POP than those women in labor ≤12 hours. Women who had a history of sphincter damage and vaginal tear were about three times [AOR = 3.2(95% CI: 1.44-7.1)] more likely to have POP than their counterparts. The women who practiced carrying/ heavy loads daily were 2.5 times more likely to develop POP than non-carriers [AOR = 2.5(95% CI: 1.2, 5.35)] (Table 5).

## Discussion

This study primarily assessed associated factors among women with POP at Public Health Hospitals of Southern Ethiopia. Accordingly, age, educational status, BMI, monthly income, higher parity, mode of delivery, duration of labor, history of sphincter damage, and carrying heavy loads were associated factors of POP.

In this study, age ≥45 years increased the risk of developing POP. Accordingly, women who were aged ≥45 years were about 5 times more likely to develop POP. This finding is supported by the study conducted in Bahr Dar [19], Gondar [16], and Rural Bangladesh [6]. The possible scientific explanation for this finding is most probably at menopause, there is a significant drop in protein content and estrogens within female reproductive tracts and supporting structures. This will result in age-related loss of muscle tissue integrity, elasticity, strength, and density, ultimately results decrease in mechanical strength and predisposes to POP [25].

This study revealed all levels of formal education were significant protective factor of POP. The odds of developing POP were 68%, and 83% lower among women of the secondary, and diploma (and above) education, respectively. This is in line with study findings reported from Bahr Dar [19] and Kersa District, Ethiopia [17]. In Ethiopia, women residing in urban areas tend to be educated, NGOs or government employees, and well aware of their health by getting information through different sources. On the other hand, in this study, about 35% cannot read & write, and about 66% of participants were rural residents who practiced tiring physical work and could not get access to health information due to lack of electricity. In rural Ethiopia in general and in the study areas in particular, hospitals are distantly located. This also contributes to the delayed treatment seeking among women with the prolapse. Most probably formally educated women are readily available for information, and timely get obstetric care including prenatal, postnatal, and delivery cares. Moreover, educated women may be more

**Table 5. Logistic regression on predictors of POP among women at Public Hospitals of Southern Ethiopia, 2020.**

| Variables | | Cases (123) | Controls (246) | COR(95%CI) | AOR(95%CI) |
|---|---|---|---|---|---|
| Age | ≤34 | 13(12.1%) | 94(87.9%) | 1 | 1 |
| | 35-44 | 35(25.2%) | 104(74.8%) | 2.57(1.29-5.13) | 2.1(0.65-19.26) |
| | ≥45 | 75(61%) | 48(39%) | 11(5.54-21.81) | **5.33(1.47-9.05)**[*] |
| Educational status | No schooling | 83(63.8%) | 47(36.2%) | 1 | 1 |
| | Primary | 18(20.5%) | 70(79.5%) | 0.14(0.8-0.27) | 0.72(0.11-1.05) |
| | Secondary | 15(18.3%) | 67(81.7%) | 0.12(0.59-02) | **0.32(0.11-0.92)**[*] |
| | Diploma[+] | 7(10.1%) | 62(89.9%) | 0.1(0.03-0.15) | **0.17(0.05-0.57)**[*] |
| BMI(kg/m²) | 18.5-25 | 96(30.5%) | 219(69.5%) | 1 | 1 |
| | <18.5 | 21(65.6%) | 11(34.4%) | 4.15(1.9-90) | **4.54(1.4-15.76)**[*] |
| | >25 | 6(27.3%) | 16(72.7%) | 1(0.4-2.50) | 1.8(0.36-8.87) |
| Monthly income /day | >$1.25 | 38(15.8%) | 203(84.8%) | 1 | 1 |
| | ≤$1.25 | 85(66.4%) | 43(33.6%) | 9.8(5.94-16.7) | **2.5(1.137-5.59)**[*] |
| Family history of POP | No | 105(32%) | 223(68%) | 1 | 1 |
| | Yes | 18(43.9%) | 23(56.1%) | 1.66(0.86-3.21) | 1.85(0.6-5.58) |
| Parity | ≤4 | 41(17.4%) | 194(82.6%) | 1 | 1 |
| | >4 | 82(61.2%) | 52(38.8%) | 7.46(4.6-12.11) | **5.2(2.2-12.55)**[*] |
| Mode of delivery at last delivery | SVD | 85(33.6%) | 168(66.4%) | 1 | 1 |
| | AVD | 34(49.3%) | 33(50.7%) | 2.04(1.18-3.51) | **4(1.55-11.63)**[*] |
| | C/SD | 4(91.8%) | 45(8.2%) | 0.18(0.06-0.51) | 1.89(0.39-7.52) |
| History of home delivery | No | 35(17.2%) | 169(82.8%) | 1 | 1 |
| | Yes | 88(53.3%) | 77(46.7%) | 9.1(5.38-15.24) | 1.44(0.55-3.8) |
| Age at last delivery | ≤40 | 83(26.2%) | 234(73.8%) | 1 | 1 |
| | >40 | 40(76.9% | 12(23.1%) | 9.4(4.70-18.77) | 2.23(0.87-4.21) |
| Duration of labor during last delivery(in hrs) | ≤12 | 38(24.8%) | 115(75.2%) | 1 | 1 |
| | 13-23 | 56(33.9%) | 109(66.1%) | 1.86(1.13-3.06) | 2.1(0.96-4.51) |
| | ≥24 | 29(56.9%) | 22(43.1%) | 4.46(2.27-8.74) | **3.3(1.12-9.7)**[*] |
| History of sphincter damage | No | 49(22.8%) | 166(77.2%) | 1 | 1 |
| | Yes | 74(48.1%) | 80(51.9%) | 3.13(2.0-4.91) | **3.2(1.44-7.1)**[*] |
| Carrying heavy objects | No | 40(18.5%) | 176(81.5%) | 1 | 1 |
| | Yes | 83(54.2%) | 70(45.8%) | 5.21(3.27 | **2.5(1.2-5.35)**[*] |

Key: Where: SPV = spontaneous vaginal delivery, AVD = assisted vaginal delivery, CS/D = cesarean section delivery, hrs = hours, AOR = adjusted odds ratios, COR = crude odds ratios.

[*] Statistically significant at P-value ≤ 0.05 in multivariable logistic regression analysis

aware of a healthy lifestyle and more open in discussing their health-related issues. Furthermore, educated women most likely can prefer jobs that are not physically laborious and can be professionally employed. This may reduce carrying heavy objects and busy household chores [9].

The underweight women were 4.5 times more likely to develop POP as compared to normal BMI values. This is consistent with the study conducted in Benchi Maji Zone [18], Bahr Dar [19], and Rural China [7]. On the other hand, the study at the Wesley Hospital reported that the women who were overweight/obese (BMI is >25 kg/m²) were riskier to develop POP [26]. The difference may be due to the difference in the lifestyle and physical work of the study participants between the two studies. The possible justification for the mechanism of POP development in underweight is most probably heavier physical workload [14], engagement in manual work even while pregnant or shortly after delivery, and poor nutrition [3] among Ethiopian women.

This result claimed to earn a daily income ≤ 1.25$ is a risk factor of POP. In most of the studies, the association of income level with POP was not tested, as it was stayed neglected. In this study, out of 104 women delayed for the treatment, 67.48% (83) did not attend formal school and 69.1% (85) of them earn less than $1.25/ day. Most probably, women with higher income can get better health services, timely seek treatments, can prefer jobs, and can hire someone to help them.

In this study, women with parity ≥ 5 were at risk of developing POP as compared to their counterparts. This is comparable with the reports of Bahr Dar [19], Gondar [16], China [7], and Bangladesh [6] study. The possible scientific explanation for this finding is more probably higher parity can weaken pelvic floor structures because of repetitive exertion of pressure during pregnancy and forceful uterine contraction during delivery as well [19]. In a recent study, the mean parity is higher (5.45). This implies the low utilization of family planning in rural Ethiopia despite available family planning by governments and NGOs. Low family planning utilization is most probably due to women in rural Ethiopia being characterized by low education, low access to health informations, beliefs and cultures that encourages the women to have many children [27]. In the study areas, having many children is considered as proud, dignity, and ultimate blessings from the Creator. Moreover, in the study areas, deciding the number of the children to have is exclusively the right of the husbands, women have no right to decide.

This study revealed that assisted vaginal delivery increases the risk to develop POP. This is consistent with a recent cohort study in Baltimore, Maryland, that reported both spontaneous and assisted vaginal deliveries were associated with a significantly higher hazard of developing POP than those delivered via cesarean delivery [28]. Besides, the finding of our study was supported by other studies [11, 29]. The possible explanation for the mechanism of POP development following spontaneous and assisted vaginal delivery is believed to result from structural disruption due to overstretching, compression, and avulsions during childbirth, and/or secondary to denervation injury to the pelvic floor muscles [30]. Both forceps and vacuum delivery appear to have a higher risk to injure pelvic floor structures [31].

This study revealed that labor duration ≥ 24 hours is risky to develop POP. More probably, Ethiopian women are prone to prolonged labor because of poor antenatal care [32], teenage marriage [33] unskilled birth attendants [32], poor road construction (especially in rural areas), inadequate ambulance services, and distant availability of general and referral hospitals for operative deliveries.

In this study, predisposition to sphincter damage due to episiotomy and vaginal tear was found to be risk factor of POP. This finding is supported by the study done at Bahr Dar [19] and Kersa District [17], Ethiopia. This owing to an injury to neuro-vasculature, disruption of the muscles, perineal membrane, ligaments, and other connective tissues of the pelvic floor that support and maintain female pelvic structures in an anatomical position [34]. However, an episiotomy is very important to ease delivery and prevent the worst consequences that could happen if the episiotomy measures would not be taken.

The odds of having POP were higher in women who carry heavy loads daily than their counterparts. This is consistent with a study in Tanzania [5], Gondar [16], and Bahr Dar [19]. This indicates that women in Ethiopia engaged in hard physical work including, fetching water, farming, carrying loads over a long distance, and carrying out household chores. In Ethiopia, women carry wood, fetch water, and carry/uplift other heavy loads over long distances while they are pregnant, postnatal state, and in nursing.

The average delay of the women for the treatments since the onset of the symptoms was 36.41 ± 3.94 months. It is very lower than similar finding from Gondar, Ethiopia, (85.8 ± 8.2 months) [22]. This discrepancy is due to that in a recent study the patients with prolapse

visited the hospitals for treatments by themselves. Meanwhile, in the study of Gondar, Ethiopia, there was a community-based screening and campaign for searching the cases of prolapse, in this case, many cases of prolapse may be surveyed for the treatments and several cases with longer duration may contribute to the high value of estimates because in the community we can find a woman who stayed for a longer period by hiding the problem.

Out of 123 women with POP, most (84.5%5) of the women arrived at hospitals with advanced stages of the prolapse. This is because they had completely hidden the problem, and they had shown it to their close relative when the condition got unbearable. As a result, all of them were treated with vaginal hysterectomies. This finding is supported by a study finding reported from Jimma, Ethiopia, that claimed about 81% of the treatments performed for the prolapses were vaginal hysterectomies. In our study, among 104 women delayed for the treatment of the prolapse, 76%, and 43.9% of them claimed lack of money, and social stigma as the main reasons for the delay, respectively.

As a strength, this study is a multi-center study and included more variables, which had not been assessed so far. In addition, testing the income variable for the association and we found it the factor that shows a significant association. To reduce recall bias, close relatives or husbands (special controls) of respondents have participated. As a limitation, this study was hospital-based and it lacks generalizability to the community at large. Even if we calculated sample size calculation, confidence intervals of some variables were wide. Hence, accompany power calculation could help evaluate the actual robustness of the results.

## Conclusions and recommendations

In conclusion, older age, low educational status, underweight, low income, higher parity, assisted vaginal delivery, prolonged labor, sphincter damage/vaginal tear, and carrying heavy loads were significant associated factors of POP. Most women delayed in the treatment of the prolapse, and most of them are treated with surgical interventions (hysterectomy). Most women claimed lack of money and social stigma as the reasons for the delay in the treatments. Screening older edge, expanding health education, pelvic muscle training, raising women's income, and reducing carrying/uplifting high loads are recommended. Moreover, all older age women especially those who are residing in rural areas need to be screened for POP, and the husbands/responsible bodies should help women in daily hard work activity; participate in handling heavy objects and reduce extended work time.

## Supporting information

**S1 File. English and Amharic version questionnaire.**
(PDF)

## Acknowledgments

We would like to pass our deepest gratitude to the data collectors, supervisors, and study participants for their commitment during data collection.

## Author Contributions

**Conceptualization:** Asfaw Borsamo, Mohammed Oumer, Ayanaw Worku, Yared Asmare.

**Data curation:** Asfaw Borsamo, Mohammed Oumer, Ayanaw Worku, Yared Asmare.

**Formal analysis:** Asfaw Borsamo, Mohammed Oumer.

**Funding acquisition:** Asfaw Borsamo.

**Investigation:** Asfaw Borsamo.

**Methodology:** Asfaw Borsamo, Mohammed Oumer, Ayanaw Worku, Yared Asmare.

**Project administration:** Asfaw Borsamo, Ayanaw Worku, Yared Asmare.

**Resources:** Asfaw Borsamo, Mohammed Oumer, Ayanaw Worku.

**Software:** Asfaw Borsamo, Mohammed Oumer, Ayanaw Worku, Yared Asmare.

**Supervision:** Asfaw Borsamo.

**Validation:** Asfaw Borsamo, Mohammed Oumer, Ayanaw Worku, Yared Asmare.

**Visualization:** Asfaw Borsamo, Mohammed Oumer, Ayanaw Worku, Yared Asmare.

**Writing – original draft:** Asfaw Borsamo, Yared Asmare.

**Writing – review & editing:** Asfaw Borsamo, Mohammed Oumer, Ayanaw Worku, Yared Asmare.

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
