## [Decision Letter · Decision Letter 0]

23 Mar 2022

PONE-D-21-33943Predictors of pelvic organ prolapse among patients at Public Hospitals of Southern Ethiopia: A case-control study designPLOS ONE

Dear Dr. Asfaw,

Thank you for submitting your manuscript to PLOS ONE. There was considerable division between reviewers here, but if you would like to address the comments than I am happy to send this for re-review but cannot guarantee a favourable decisionTherefore, we invite you to submit a revised version of the manuscript that addresses the points raised during the review process.

We look forward to receiving your revised manuscript.

Kind regards,

Adrian Stuart Wagg, MD

Academic Editor

PLOS ONE

Journal Requirements:

We would like to pass our deepest gratitude to the College of Medicine and Health Sciences, the University of Gondar, for funding (financial support) this research works. 

Mr. Asfaw Borsamo took award From University of Gondar with grant number 045028/2020. The website of University of Gondar is http://www.uog.edu.et/en/. The University of Gondar had no role in study design, data collection and analysis, decision to publish, or preparation of the manuscript.

Reviewers' comments:

Reviewer's Responses to Questions

**Comments to the Author**

1. Is the manuscript technically sound, and do the data support the conclusions?

Reviewer #1: Yes

Reviewer #2: Partly

2. Has the statistical analysis been performed appropriately and rigorously? 

Reviewer #1: Yes

Reviewer #2: Yes

3. Have the authors made all data underlying the findings in their manuscript fully available?

Reviewer #1: No

Reviewer #2: No

4. Is the manuscript presented in an intelligible fashion and written in standard English?

Reviewer #1: Yes

Reviewer #2: Yes

5. Review Comments to the Author

Reviewer #1: Reviewer’s report: Predictors of pelvic organ prolapse among patients at Public Hospitals of Southern Ethiopia: A case-control study design

the aim of this case control study was to examine predictors of pelvic organ prolapse patients at southern Ethiopian public hospitals

Abstract:

the conclusion is largely valid but the last sentence is perhaps somewhat speculative. Perhaps the authors might like to modify this.

Introduction

The authors note the highly variable prevalence of prolapse in studies which existed to date perhaps they might confine this detailed discussion to the discussion section and perhaps suggest some explanation as to why this is the case.

The authors note established risk factors in other studies, could they therefore suggest why a study in Ethiopia would be of value or might produce different information?

Would a study attempting to intervene on the known modifiable factors be more significant and have greater utility than reiterating what is known?

the authors make a good point about using the data with “responsible bodies” in playing a role of modification of such factors which would be a useful motivation upon which to build

methods and materials

It is good to see an accurate sample size calculation. what was the degree of imprecision in the prevalence estimate?

please give the authors describe their random sampling technique For participants and hospitals?

Was the Constipation question validated?

Who examined the women?

The authors are to be congratulated in the conduct of this study in a challenging environment, in particular the governance processes they have put in place to ensure data quality

The statistical analysis section is entirely appropriate.

I note that written informed consent was obtained from study participants, What proportion of women was illiterate and how was consent gained in this case?

Results

For what reason have the authors included the religion of the women? Was this used as a controlling variable in parity? Did religion and parity interact?

Is the proportion of women earning greater than 1200 birrs representative of the Ethiopian general population of women? Likewise, is the proportion having a home delivery typical?

Is the history of Sphincter damage and vagina tear when gained from the women reliable?

Regarding predictors, were any variables discarded on bivariate analysis? Were there any interactions? For example was high parity associated with relative poverty?

Likewise was carrying heavy loads more common in women who were of lower educational attainment or low income?

The results section is otherwise extremely well written But I should like to see some comparative analysis of the factors in table one in terms of prevalence of prolapse and say religion occupational status and ethnicity for completions sake

Can the authors tell us what the average delay in seeking treatment was for the women both as a whole and according to income status education status or employment status?

This might help to support their argument that women with high income get better health services seek timely treatments and can hire someone to help them

Discussion

the authors present their findings in a logical manner and attempts to explain their findings.

Might the authors also consider the amount of physical exertion women perform separate from carrying heavy loans as a potential mitigating factor - as the authors note this might also account for the higher prevalence of prolapse in underweight women

The authors might consider comparing their sample with the general Ethiopian population to give the reader some frame of reference for example, were the women more educated, or likely to be employed or were they wealthier than average women in Ethiopia

the high parity doesn't necessarily imply low utilization of family planning - the high parity may well be purposeful. do the authors have any evidence to suggest that women want fewer children?

The authors have considered their limitations and biases, notably the external validity of this hospital population seeking help versus the general population

to what extent were husbands likely to give reliable information on their wives?

In conclusion the authors have identified many potentially remedial factors in the development of prolapse. how likely are women to choose active treatment of their prolapse in Ethiopia. Is operative intervention financially out of reach for the majority of women? What conservative therapies are available?

This information would certainly help add context to the paper

Reviewer #2: The authors investgated in the present study the effect estmate of several known factors that predispose to pop in an ethiopian population. Despite their rigorous methodology i believe that the lack of an appropriate sample size (which is denoted by the large confidence intervals that are provided) is the main reason for rejection of this article.

6. PLOS authors have the option to publish the peer review history of their article (what does this mean?). If published, this will include your full peer review and any attached files.

Reviewer #1: No

Reviewer #2: No

---

## [Author Response · Author response to Decision Letter 0]

26 May 2022

Responses to Reviewers: 

Corrections/revisions for Reviewer 1

Abstract:

1. The conclusion is largely valid but the last sentence is perhaps somewhat speculative. Perhaps the authors might like to modify this. 

Dear reviewer, we found this comment is relevant and we have corrected/revised as your request. Thank you very much for your suggestions and recommendations.

Introduction:

2. The authors note the highly variable prevalence of prolapse in studies which existed to date perhaps they might confine this detailed discussion to the discussion section and perhaps suggest some explanation as to why this is the case. 

Dear reviewer, the objective of the study is to determine predictors of the pelvic organ prolapse. This why we preferred to discuss in detail about the prevalence of the pelvic organ prolapse in introduction part.

3. The authors note established risk factors in other studies, could they therefore suggest why a study in Ethiopia would be of value or might produce different information? 

Would a study attempting to intervene on the known modifiable factors be more significant and have greater utility than reiterating what is known? the authors make a good point about using the data with “responsible bodies” in playing a role of modification of such factors which would be a useful motivation upon which to build

Dear reviewer, the most risk factors of the prolapse varies greater from the community to community because of different socioeconomic status, health system development level, and culture or beliefs. According to the literatures, so many factors (over 60) have association with the prolapse. Policy makers and health managers would not intervene all risk factors of the prolapse. This study was conducted with a case control study design to identify which factors are significantly affecting the prolapse in the study areas. Hence, managers and policy makers of the study areas possibly can work on identified factors to modify.

Methods and materials

4. It is good to see an accurate sample size calculation. what was the degree of imprecision in the prevalence estimate? 

Dear reviewer, previously, we had written the sample size calculation in short and summarized way. Now, on the revised manuscript, we have explicitly explained how the sample size was calculated as following:

In this study, sample size was calculated by using Epi info version 7 software. For unmatched case-control study design, this software calculates by using double proportion formula [n= (p1q1 + p2q2) (f(�,�)) / ((p1 - p2)², n= number of sample size in each group, � = type I error, � = type II error]. To calculate sample size, we used proportions from the similar study conducted in Bahr Dar City, Ethiopia, as following: (see Table 1)

Risk actors % of case with exposure % of control with exposure Power CI (%) AOR Sample size

Age >44 71.8 27.1 80 95 6.8 55

Formal education 85 58 80 95 4.3 111

Parity/delivery 64.5 26.67 80 95 4.5 69

Sphincter damage 15.2 2 80 95 8 173

Carry heavy object 67.4 40 80 95 3.1 131

Family history of POP 22.6 6.2 80 95 4.9 176

Underweight 34.5 14 80 95 3.2 164

Delivery assisted by non-health personnel 63.4 39.1 80 95 2.6 167

We got maximum sample size with variable of ‘family history of POP’ which is 173, after considering the design effect of two, and a 5% non-response rate, the total sample size became 369 patients (123 cases and 246 controls: in 1 to 2 cases to controls ratio).

5. Please give the authors describe their random sampling technique for participants and hospitals? 

Dear reviewer, among 14 governmental general and referral hospitals in SNNPR, seven hospitals were selected by simple random method (lottery method). All patients with a pelvic organ prolapse visiting gynecologic OPD during data collection period were included as cases. To select controls, we used systematic random sampling method, depending on gynecologic patient flow of each hospital. Hence, the K value for each hospital varies (Figure 1) 



















N/E/M/M/= Negest Elleni Mohammed Memoral, HU = Hawassa University, Ca=cases, Co=controls

Figure 1: Sampling procedure

5. Was the Constipation question validated? 

Dear reviewer, we considered constipation lasting longer than three months as a significant enough to contribute for the development of the prolapse. Many other studies also agree with this statement.

6. Who examined the women? 

Gynecologists performed physical examination, history taking, and diagnosis as already explained in the manuscript.

7. The authors are to be congratulated in the conduct of this study in a challenging environment, in particular the governance processes they have put in place to ensure data quality. 

The statistical analysis section is entirely appropriate. 

Dear reviewer, we thank you very much for understanding our efforts and challenging environment that we had faced.

8. I note that written informed consent was obtained from study participants, What proportion of women was illiterate and how was consent gained in this case? 

Dear reviewer, thank you for your important comments, a written informed consent was obtained directly from study participants who can read and write. For those unable to read and write, first the consent was clearly read to her and then her immediate/ significant family member (like husband or son, etc) will sign it. Even she has not family member, the consent was clearly read to her and then her fingerprints were taken as signature.

Results

9. For what reason have the authors included the religion of the women? Was this used as a controlling variable in parity? Did religion and parity interact? 

Dear reviewer, religion of the respondents first included as sociodemographic characteristic for the descriptive purpose. Later, its association with the prolapse checked on multivariable logistic regression and it did not show association. We have checked that no interaction of the religion and parity.

10. Is the proportion of women earning greater than 1200 birrs representative of the Ethiopian general population of women? Likewise, is the proportion having a home delivery typical? 

Dear reviewer, this study categorizes the income by using World Bank cut off point of poverty line, as operationally defined in the main manuscript. According to World Bank definition of the poverty line, earning below $1.25/day(1200 birr/month) is considered below poverty line. In our study, about 69% of the women with prolapse earn less than $1.25/day; about 17.47% of women from the control earn less than $1.25/day. In our study, 35.58% of the women with prolapse had at least one history of home delivery and about 30% of women from controls had at least one history of home delivery in their life. We don’t think shortage of the income affects home delivery in Ethiopian because medical services related with obstetrics is for free. Rather, home delivery is related to cultures, awareness, and attitude

11. Is the history of Sphincter damage and vagina tear when gained from the women reliable?

Dear reviewer, yes, it is reliable because they never forget history that they had episiotomy or vagina tear. Moreover, their husbands had helped during the interview for some. 

12. Regarding predictors, were any variables discarded on bivariate analysis? Were there any interactions? For example was high parity associated with relative poverty?

Variables such as ethnicity, marital status, religion, residency, home delivery, duration of time to return to work after delivery, delivery interval, age at marriage, age at first delivery, age at last delivery, chronic cough and chronic constipation were discarded on bivariate logistic analysis.

13. Likewise was carrying heavy loads more common in women who were of lower educational attainment or low income? 

Dear reviewer, yes, carrying heavy loads more common in women who were of lower educational attainment or low income.

14. The results section is otherwise extremely well written But I should like to see some comparative analysis of the factors in table one in terms of prevalence of prolapse and say religion occupational status and ethnicity for completions sake. 

Dear reviewer, Table 2 shows these very clearly, and one can easily calculate from the table. We done in such way to reduce space and explain in summarized way..

15. Can the authors tell us what the average delay in seeking treatment was for the women both as a whole and according to income status education status or employment status? 

This might help to support their argument that women with high income get better health services seek timely treatments and can hire someone to help them

Dear reviewer, we found your concerns as very important and we have included it also in the main manuscript.

The average delay of the women for the treatments since onset of the symptoms was 36.41 ± 3.94 months. Out of 123 women with the prolapse, 84.55% delayed for the treatments and only surgical interventions done as treatments. Among 104 women delayed for the treatment, 76% of them claimed lack of money as main reason for the delay. Out of 104 women delayed for the treatment, 67.48% (83) did not attend formal school and 69.1% (85) of them earn less than $1.25/ day.

Discussion

16. The authors present their findings in a logical manner and attempts to explain their findings.

Might the authors also consider the amount of physical exertion women perform separate from carrying heavy loans as a potential mitigating factor - as the authors note this might also account for the higher prevalence of prolapse in underweight women. 

Of course, the reviewer is right, but for us it was difficult to measure amount of exertion force other than carrying heavy loads. We considered all type of heavy physical forces as carrying heavy loads.

17. The authors might consider comparing their sample with the general Ethiopian population to give the reader some frame of reference for example, were the women more educated, or likely to be employed or were they wealthier than average women in Ethiopia. 

Dear reviewer, we found this a very important comment and we incorporated this comment into the manuscript. 

18. The high parity doesn't necessarily imply low utilization of family planning - the high parity may well be purposeful. do the authors have any evidence to suggest that women want fewer children? 

Dear reviewer, we appreciate you for rising important idea, and we have made modifications accordingly. 

In Ethiopia, there is low utilization of family planning especially among rural residents. This because women in rural Ethiopia are characterized by low education, low access to health informations, beliefs and cultures that encourages the women to have many children. In the study areas, having many children is considered as proud, dignity, and ultimate blessings from the Creator. Moreover, in the study areas, deciding the number of the children to have exclusively the right of the husbands, women have no right to decide.

19. The authors have considered their limitations and biases, notably the external validity of this hospital population seeking help versus the general population

to what extent were husbands likely to give reliable information on their wives?

During the data collection, we found that husbands’ helps were very important especially when women had forgotten some events in their life history. They knew almost everything about their wives.

20. In conclusion the authors have identified many potentially remedial factors in the development of prolapse. how likely are women to choose active treatment of their prolapse in Ethiopia. 

Dear reviewer, thank you for your comment and we included it into main manuscript accordingly.

Most of the women arrived at hospital at advanced stage of the prolapse. They completely had hidden the problem and they had showed to their close relative when the condition got unbearable. As a result no options than surgical treatments. During the data collection, out of 123 women with prolapse 104 (84.55%) were surgically repaired (vaginal hysterectomies).

21. Is operative intervention financially out of reach for the majority of women? 

Dear reviewer, thank you for your comment and we included it into main manuscript accordingly

Among 104 women delayed for the treatment, 76% of them claimed lack of money as main reason for the delay. Most of the women with prolapse come to hospital after they had got some financial support from some people who understood their conditions. 

22. What conservative therapies are available? This information would certainly help add context to the paper

Dear reviewer, rest of the women with less advanced stages ( stage II) were treated with conservative means of treatments like lifestyle and behavior modification, physical therapy for the pelvic floor, and using vaginal device (Pessary)

Corrections/revisions for Reviewer 2

1. The authors investgated in the present study the effect estimate of several known factors that predispose to pop in an Ethiopian population. Despite their rigorous methodology i believe that the lack of an appropriate sample size (which is denoted by the large confidence intervals that are provided) is the main reason for rejection of this article. 

Dear reviewer, we thank you very much for your concern. The most risk factors of the prolapse vary greater from the community to community because of different socioeconomic status, health system development level, and culture or beliefs. According to the literatures, so many factors (over 60) have association with the prolapse. Policy makers and health managers would not intervene all risk factors of the prolapse. This study was conducted with a case control study design to measure which factors are significantly affecting the prolapse in the study areas. Hence, managers and policy makers of the study areas possibly can work on identified factors to modify.

Regarding sample size calculation,

Dear reviewer, our sample size very high for case-control study and we calculated and got the value according scientific rule. Everything in this study is scientifically randomized.

Dear reviewer, previously, we had written the sample size calculation in short and summarized way. Now, on the revised manuscript, we have explicitly explained how the sample size was calculated as following:

In this study, sample size was calculated by using Epi info version 7 software. For unmatched case-control study design, this software calculates by using double proportion formula or power approach [n= (p1q1 + p2q2) (f(�,�)) / ((p1 - p2)², n= number of sample size in each group, � = type I error, � = type II error]. To calculate sample size, we used proportions from the similar study conducted in Bahr Dar City, Ethiopia, as following: (see Table 1)

We got maximum sample size with variable of ‘family history of POP’ which is 173, after considering the design effect of two, and a 5% non-response rate, the total sample size became 369 patients (123 cases and 246 controls: in 1 to 2 cases to controls ratio).

Table 1: sample size calculation

Risk actors % of case with exposure % of control with exposure Power CI (%) AOR Sample size

Age >44 71.8 27.1 80 95 6.8 55

Formal education 85 58 80 95 4.3 111

Parity/delivery 64.5 26.67 80 95 4.5 69

Sphincter damage 15.2 2 80 95 8 173

Carry heavy object 67.4 40 80 95 3.1 131

Family history of POP 22.6 6.2 80 95 4.9 176

Underweight 34.5 14 80 95 3.2 164

Delivery assisted by non-health personnel 63.4 39.1 80 95 2.6 167

---

## [Decision Letter · Decision Letter 1]

16 Aug 2022

PONE-D-21-33943R1Predictors of pelvic organ prolapse among patients at Public Hospitals of Southern Ethiopia: A case-control study designPLOS ONE

Dear Dr. Asfaw,

Thank you for submitting your manuscript to PLOS ONE. After careful consideration, we feel that there is a little more work to do to fully meet PLOS ONE’s publication criteria as it currently stands. Therefore, we invite you to submit a revised version of the manuscript that addresses the points raised during the review process.

ACADEMIC EDITOR: Please insert comments here and delete this placeholder text when finished. Be sure to:Reviewer 2 makes useful points about prediction - a term you should probably avoid unless you would like to commit to further analytical models - it may be simpler to avoid the term

We look forward to receiving your revised manuscript.

Kind regards,

Adrian Stuart Wagg, MD

Academic Editor

PLOS ONE

Journal Requirements:

Reviewers' comments:

Reviewer's Responses to Questions

**Comments to the Author**

1. If the authors have adequately addressed your comments raised in a previous round of review and you feel that this manuscript is now acceptable for publication, you may indicate that here to bypass the “Comments to the Author” section, enter your conflict of interest statement in the “Confidential to Editor” section, and submit your "Accept" recommendation.

Reviewer #1: All comments have been addressed

Reviewer #2: All comments have been addressed

2. Is the manuscript technically sound, and do the data support the conclusions?

Reviewer #1: Yes

Reviewer #2: No

3. Has the statistical analysis been performed appropriately and rigorously? 

Reviewer #1: Yes

Reviewer #2: Yes

4. Have the authors made all data underlying the findings in their manuscript fully available?

Reviewer #1: Yes

Reviewer #2: Yes

5. Is the manuscript presented in an intelligible fashion and written in standard English?

Reviewer #1: Yes

Reviewer #2: Yes

6. Review Comments to the Author

Reviewer #1: The authors have prepared an excellent response to the reviewers comments and have added in information which makes this an informative paper. The abstract might benefit from adding into the results the delay results – this would then justify their conclusion about awareness raising etc

Id like to see the consent response added into the paper – this is useful for those conducting research in areas with high levels of illiteracy

Reviewer #2: The authors investigated in the present study the impact of several factors on the actual incidence of POP. The following comments are necessary to be addressed prior to its acceptance.

1) If the predictive value of these factors is to be investigated a ROC and AUC analysis along with sensitivity and specificity of individual factors or the whole logistic regression should be provided. Otherwise the study should be targeted only around the risk factors and not their predictive role. If the predictive role is to be investigated the Hosmer Lemeshow goodness of fit of the logistic regression should be reported.

2) Despite having performed a sample size calculation, the confidence intervals that the authors report are rather wide; hence, an accompanying power calculation of their findings could help evaluate the actual robustness of their results and comment on them in the discussion.

7. PLOS authors have the option to publish the peer review history of their article (what does this mean?). If published, this will include your full peer review and any attached files.

Reviewer #1: No

Reviewer #2: **Yes: **Vasilios Pergialiotis

---

## [Author Response · Author response to Decision Letter 1]

27 Sep 2022

Response Letter 

Authors: 

1. Mr. Asfaw Borsamo 

2. Mr. Mohammed Oumer 

3. Mr. Ayanaw Worku 

4. Mr. Yared Asmare 

Editorial-in-Chief of the PLOS ONE JOURNAL

 October 03, 2022

Dear Dr. (Professor) Adrian Stuart Wagg,

 Please find enclosed a revised manuscript entitled “Associated factors of pelvic organ prolapse among patients at Public Hospitals of Southern Ethiopia: A case-control study design’’ with the kind request to consider it for publication in the Plos One Journal.

In the first place, we would like to thank the Esteemed Dr. Adrian Stuart Wagg and Reviewers for your kind and lesson giving comments, create an opportunity to improve our work. We have accepted the comments, and we made modifications accordingly. For the concerns that were raised by the esteemed reviewers, we have given explanations. We have checked that our references list is complete and correct.

Journal requirement: We have checked that our references list is complete and correct.

Responses to Reviewers: 

Corrections/revisions for Reviewer 1

1. The authors have prepared an excellent response to the reviewers’ comments and have added in information which makes this an informative paper. The abstract might benefit from adding into the results the delay results – this would then justify their conclusion about awareness raising etc.

Dear reviewer, we would like to thank for appreciating a previous response and we found the recent comment is also relevant. Hence, we have corrected/revised as your request. Thank you very much for your suggestions and recommendations. Your comments are making our work complete. 

2. Id like to see the consent response added into the paper – this is useful for those conducting research in areas with high levels of illiteracy 

Dear reviewer, we used the consent form as following:

 Consent form 

English version

I have read or it was read to me (for those who cannot read and write) and understood well the condition stated above and I can withdraw from the study at any time and I understand that there is no risk of participating and no incentive to be given when I participate in the study. Therefore, I am willing to participate in the study. Signature___________________ Date____________ 

 Thank you so much! 

 Amharic version

የስምምነት ፍቃድ 

ከላይ የተጻፈውን መረጃ በደንብ አንብቤ ተረድቸው አለሁ፡፡ በጥናቱ ላይ መሳተፍ በኔም ሆነ በልጀ ላይ ምንም አይነት ችግር እንደማያደርስ ፤ የተለየ ጥቅማጥቅምም እንደለለው እንዲሁም በማነኛውም ሰአት ካልተመቸኝ የማቋረጥ መብት እንዳለኝ ተነግሮኛል፡፡ በመሆኑም በጥናቱ ላይ ለመሳተፍ ዝግጁ/ፍቃደኛ ነኝ ፡፡ 

 ፊርማ --------------------- ቀን --------------------------------

 ስለተሳትፎዎ እናመሰግን አለን !!! 

Corrections/revisions for Reviewer 2

Thank you very much for your suggestions and recommendations.

1. If the predictive value of these factors is to be investigated a ROC and AUC analysis along with sensitivity and specificity of individual factors or the whole logistic regression should be provided. Otherwise the study should be targeted only around the risk factors and not their predictive role. If the predictive role is to be investigated the Hosmer Lemeshow goodness of fit of the logistic regression should be reported.

Dear reviewer, thank you so much for your comment. We have modified our manuscript; we have replaced the term “predictors” with term “associated factors” throughout the manuscript. Mostly in public health research, researchers prefer the term “determinant” for cohort studies, “predictors” for case-control studies and retrospective cohort studies, and “associated factors” for cross-sectional studies. Similarly, we used the term “predictor” simply to show the level of term choice. So, thank you the Esteemed Academic Editor and Reviewer, now use the term “associated factors” Thank you very much for your suggestions and recommendations.

2. Despite having performed a sample size calculation, the confidence intervals that the authors report are ather wide; hence, an accompanying power calculation of their findings could help evaluate the actual robustness of their results and comment on them in the discussion. 

Dear reviewer, we have incorporated per your request in the discussion. Thank you very much.

---

## [Decision Letter · Decision Letter 2]

17 Nov 2022

Associated factors of pelvic organ prolapse among patients at Public Hospitals of Southern Ethiopia: A case-control study design

PONE-D-21-33943R2

Dear Dr. Asfaw,

We’re pleased to inform you that your manuscript has been judged scientifically suitable for publication and will be formally accepted for publication once it meets all outstanding technical requirements.

Kind regards,

Joseph Donlan

Staff Editor

PLOS ONE

Additional Editor Comments (optional):

Reviewers' comments:

Reviewer's Responses to Questions

**Comments to the Author**

1. If the authors have adequately addressed your comments raised in a previous round of review and you feel that this manuscript is now acceptable for publication, you may indicate that here to bypass the “Comments to the Author” section, enter your conflict of interest statement in the “Confidential to Editor” section, and submit your "Accept" recommendation.

Reviewer #1: All comments have been addressed

2. Is the manuscript technically sound, and do the data support the conclusions?

Reviewer #1: Yes

3. Has the statistical analysis been performed appropriately and rigorously? 

Reviewer #1: Yes

4. Have the authors made all data underlying the findings in their manuscript fully available?

Reviewer #1: Yes

5. Is the manuscript presented in an intelligible fashion and written in standard English?

Reviewer #1: No

6. Review Comments to the Author

Reviewer #1: thank you for your revisions. This has improved the paper. I think you made the right choice in not complicating your analysis which would have meant a lot of reworking

7. PLOS authors have the option to publish the peer review history of their article (what does this mean?). If published, this will include your full peer review and any attached files.

Reviewer #1: No

---

## [Editor Report · Acceptance letter]

9 Jan 2023

PONE-D-21-33943R2 

Associated factors of pelvic organ prolapse among patients at Public Hospitals of Southern Ethiopia: A case-control study design 

Dear Dr. Asfaw:

I'm pleased to inform you that your manuscript has been deemed suitable for publication in PLOS ONE. Congratulations! Your manuscript is now with our production department. 

Kind regards, 

on behalf of

Dr Joseph Donlan 

Staff Editor

PLOS ONE